# Evolution of strain diversity and virulence factor repertoire in pediatric *Staphylococcus aureus* isolates

Margaret Free[1,2]*, Nicole Soper[1,2], James C. Slaughter[3], Andries Feder[4,5], Colleen Bianco[4,5], Ahmed M. Moustafa[5,6], Paul Planet[4,5,7], C. Buddy Creech[1,2], Isaac Thomsen[1,2]

1 Department of Pediatrics, Division of Infectious Diseases, Vanderbilt University Medical Center, Nashville, Tennessee, United States of America, 2 Vanderbilt Vaccine Research Program, Vanderbilt University Medical Center, Nashville, Tennessee, United States of America, 3 Department of Biostatistics, Vanderbilt University School of Medicine, Nashville, Tennessee, United States of America, 4 Division of Pediatric Infectious Diseases, Children's Hospital of Philadelphia, Philadelphia, Pennsylvania, United States of America, 5 Department of Pediatrics, University of Pennsylvania, Philadelphia, Pennsylvania, United States of America, 6 Division of Gastroenterology, Hepatology, and Nutrition, Children's Hospital of Philadelphia, Philadelphia, Pennsylvania, United States of America, 7 Comparative Genomics, American Museum of Natural History, New York, New York, United States of America

* Margaret.Free@SanfordHealth.org, margaretfree05@gmail.com

## Abstract

### Background

Invasive *Staphylococcus aureus* infections cause high morbidity and mortality in children and adults. With rising antimicrobial resistance, optimal prevention strategies and novel therapeutics are needed. As an effective vaccine remains elusive, characterization of invasive isolates over time is required to identify determinants of invasive infection.

### Methods

*S. aureus* isolates recovered from children with invasive infection and those with colonization were obtained. Isolates were examined by whole genome sequencing to evaluate gene repertoire, sequence type, clonal complex, and phylogenetic characterization, and isolate characteristics were correlated to clinical data.

### Results

118 children with invasive *S. aureus* infections were enrolled; 56% of infections were caused by methicillin-susceptible *S. aureus* (MSSA). Methicillin-resistance (MRSA) was associated with increased inflammation, though clinical outcomes of MRSA vs MSSA did not differ. Colonization isolates exhibited higher sequence type diversity than invasive isolates. Nine distinct clonal complexes (CC) were identified among all isolates; CC8 and CC5 were associated with higher clinical severity scores.

**Data availability statement:** All relevant data are within the paper, its Supporting Information files, and from the PubMLST database (https://pubmlst.org/bigsdb?db=pubmlst_saureus_isolates).

**Funding:** This work was supported through the NIH/NIAID grants 5R01AI139172-02 (Thomsen PI) and 5T32AI095202 (Childhood Infections Research Program Training Grant) as well as internal funding from Vanderbilt University Medical Center: the David T Karzon Award. The David T Karzon Award is a donor award and does not have a grant number or reference tag. The funders had no role in study design, data collection and interpretation, or submission of the work for publication. The data used in Fig 3 were collected through the MENDEL high performance computing (HPC) cluster at the American Museum of Natural History. This HPC cluster was developed with National Science Foundation (NSF) Campus Cyberinfrastructure support through Award#1925590. We thank the VUMC Clinical Microbiology Laboratory, the Vanderbilt Vaccine Research Program Laboratory (VVRP), and the Vanderbilt Technologies for Advanced Genomics (VANTAGE) Core. VANTAGE is supported in part by Clinical and Translational Science Award Grant 5UL1 RR024975-03, Vanderbilt Ingram Cancer Center Grant P30 CA68485, Vanderbilt Vision Center Grant P30 EY08126, and National Institutes of Health/National Center for Research Resources Grant G20 RR030956.

**Competing interests:** C.B.C. is a consultant for Pfizer, Moderna, GlaxoSmithKline (GSK), Sanofi, and Cowen Investments and receives royalties from UpToDate. C.B.C. and I.T. have a patent on Staphylococcus aureus monoclonal antibody US10981979B2. This does not alter our adherence to PLOS ONE policies on sharing data and materials.

Accessory gene regulator locus type 1, Panton-Valentine Leukocidin, and arginine catabolic mobile element declined over time. Staphylokinase and leukocidin ED were associated with invasive infection, while enterotoxin B was more frequent in colonizing isolates.

## Conclusions

We observed a significant expansion in sequence type diversity among invasive clinical isolates over 12 years with the emergence of newly invasive clones in recent years. The presence of staphylokinase and LukED were associated with invasive infection over time. These findings provide insights into the pathogenesis of invasive *S. aureus* and may provide putative targets for immunologic approaches to prevention.

---

## Introduction

*Staphylococcus aureus* is the most common bacterial pathogen isolated in children with bacteremia and musculoskeletal infections [1] and represents the most commonly identified bacterial pathogen in fatal infections globally [2]. Although *S. aureus* is a commensal, it can breach normal host immune defenses, leading to invasive infections such as bacteremia, bacterial arthritis, myositis, osteomyelitis, and pneumonia. It remains unclear why some otherwise healthy children succumb to severe, invasive infections.

The constant evolution of *S. aureus* is illustrated in the epidemic rise, and now gradual decline, of community-associated MRSA (CA-MRSA). Until the late 1980s, MRSA infections were almost exclusively seen in patients with nosocomial acquisition (e.g., patients with prolonged hospitalizations or those residing in long-term care facilities) [3]. In 1997, four previously healthy children died of MRSA infections, and, over the subsequent decade, CA-MRSA caused most invasive staphylococcal infections in children, prompting the common inclusion of anti-MRSA antibiotics in empiric treatment of presumed staphylococcal infections [4]. Over the last decade, a relative increase in MSSA has been observed [5], and MSSA now represents the majority of invasive *S. aureus* infections in children in the United States [6–8].

Despite decades of inquiry into *S. aureus* genotypes, surface proteins, and extracellular toxins, the bacterial factors that contribute to disease progression in humans have proven difficult to establish conclusively; as a result, an effective staphylococcal vaccine remains elusive. Many recent and current vaccine candidates target antigens that have appeared critical to pathogenesis, only to see many of these virulence determinants recede from circulating invasive strains [9,10]. Therefore, the characterization of clinically relevant *S. aureus* strains over time and identification of those factors that remain associated with invasive infection are necessary to find effective targets of intervention.

In this study, we used whole genome sequencing to evaluate *S. aureus* virulence genes, sequence types, clonal complexes, and genetic relatedness of pediatric

invasive *S. aureus* isolates over 12 years (2010–2022) and pediatric colonizing isolates during 2 distinct time periods (2004 and 2021–2022). Among invasive isolates, we examined clinical data to determine if specific bacterial genotypes correlate with severity of invasive infection.

## Methods

### Study population

All patients were enrolled at a single center (Vanderbilt University Medical Center [VUMC]). Patients aged 6 months – 18 years who were hospitalized with invasive, culture-proven *S. aureus* infections were eligible for the study. Patients with severe immunocompromise were excluded (example: a patient with relapsed acute myelogenous leukemia on chemotherapy with secondary neutropenia was excluded, a patient with cystic fibrosis and osteomyelitis was included, a patient with acute lymphoblastic leukemia in remission with a normal neutrophil count and pyomyositis was included).

### Patient consent statement

The study was approved by the Vanderbilt University Medical Center Human Subjects Protection Program. Prior to the conduct of study related procedures, written informed consent was obtained from the legal guardian of patients under age 18 years with written assent obtained from patients when developmentally appropriate, and written informed consent was obtained from patients 18 years of age. The recruitment period began June 17, 2010 and ended May 19, 2022.

Colonization isolates were obtained in 2004 from children between the ages of 2 weeks and 21 years presenting for health maintenance visits in the outpatient setting [11] under a separate IRB-approved study. Isolate information from colonization isolates obtained in 2004 was accessed in October 2022, and personally identifiable information was not accessed in the analysis of the samples. From July 2021 – July 2022, colonization isolates were obtained on a quarterly basis from anonymized infants admitted to the neonatal intensive care unit at our institution, per the standard of care. All participants were enrolled from a single center (VUMC).

### Definitions

Invasive staphylococcal infection was defined by growth of *S. aureus* in culture from a normally sterile site in patients with clinical phenotypes consistent with invasive infections. The invasive infection phenotypes included uncomplicated bacteremia, complicated bacteremia, osteomyelitis, bacterial arthritis, pyomyositis, endocarditis, and pneumonia (pneumonia required growth of *S. aureus* from pleural fluid or bronchoscopy fluid for enrollment).

Isolates from patients with central line associated blood stream infections (CLABSI), indwelling catheters (e.g., urinary catheters or peritoneal dialysis catheters), or surgical site infections with hardware or allograft in place (n = 13) were included in the clinical correlation analysis but not sent for whole genome sequencing.

To broadly estimate *S. aureus* disease severity, a clinical severity score was assigned for each patient, ranging from 0 to 8, where one point was given for each of the following: patient death, hospital stay >14 days, ICU admission, thrombosis/septic emboli, prolonged bacteremia (defined as positive blood culture >72 hours after appropriate antimicrobial therapy), CRP > 150 mg/L, peak absolute neutrophil count (ANC) >20,000/mcL, nadir ANC < 1,500/mcL, and nadir platelet count <100,000/mcL.

### Isolation of DNA and molecular typing

*S. aureus* isolates were obtained from the VUMC Clinical Microbiology Laboratory, plated onto tryptic soy agar (with 5% sheep blood), and incubated at 37˚C with 5% $CO_2$ overnight. Antimicrobial resistance data were collected from the VUMC Clinical Microbiology Laboratory. Genomic DNA (gDNA) was extracted using a Qiagen DNA easy ultraclean microbial kit and used as template DNA. For ACME typing and agr typing, DNA was amplified using polymerase chain reaction and was visualized with agarose gel electrophoresis [12–15].

For phylogenetic analysis, multilocus sequence typing, and assessment for presence of all other virulence genes, whole genome sequencing was performed by Vanderbilt Technologies for Advanced Genomics (VANTAGE) Core (Nashville, TN, USA). DNA samples were submitted to VANTAGE for library construction and sequencing. The quantity of DNA was determined using a Picogreen assay and the samples were normalized to 50–100 ng per sample. Libraries were prepared with the Twist Biosciences Kit (P/N 104206) according to manufacturer's instructions. The libraries were amplified using PCR with indexed primers to introduce unique barcodes for sample identification and to amplify the target DNA fragments. The samples were purified using beads to remove unused primers, dNTPs, and other reaction components. The final libraries were quantified using qPCR and a fluorometric quantification assay to normalize and pool for sequencing. The libraries were sequenced using the NovaSeq 6000 (Illumina) using 150 base pair paired-end reads targeting 4 million reads per sample. Real Time Analysis Software (v2.4.11; Illumina) was used for base calling and quality determination. The file deliverable was a demultiplexed FASTQ file containing the pass-filter (PF) reads. FASTQ files were imported into Geneious Prime 2022.2.2 to determine the presence of 20 virulence genes. Raw FASTQ files were imported and mapped to reference strains (GenBank accession numbers CP014444 and CP090874). Consensus sequences were generated and submitted to PubMLST for multilocus sequence typing (MLST). MLST and clonal complex identification were performed by PubMLST (an open-access curated database [16]), eBURST algorithms, and ID Genomics (Seattle, WA). All CC8 genomes were evaluated for the presence of USA300 diagnostic alleles that were previously identified [17,18]. Consensus sequences with MLST assignments remain publicly available via PubMLST.

A phylogenetic tree was constructed. The phylogeny was rooted to a *Staphylococcus argenteus* genome from our collection which was subsequently pruned for visualization purposes. A maximum likelihood tree was built using the Cladebreaker pipeline (https://github.com/andriesfeder/cladbreaker). A maximum likelihood tree was constructed for 497 genomes; 230 genomes from our collection and 267 assembled genomes available on GenBank [19], chosen using the topgenome (-t) feature of WhatsGNU [20] with 3 top genomes specificized. The genomes from our collection were processed using the bactopia pipeline v2.2.0 [21], and *de novo* assembly was completed using Shovill v1.1.0 (https://github.com/tseemann/shovill). Genome annotation was completed using Prokka v1.14.6 [22], and a pangenome alignment produced by Roary v3.13.0 [23] was used to infer an initial phylogenetic tree in RAxML v8.2.9 [24] using GTR substitution model [25] accounting for among-site rate heterogeneity using the Γ distribution and four rate categories (GTRGAMMA model [25]) for 100 individual searches with maximum parsimony random-addition starting trees. Node support was evaluated with 100 nonparametric bootstrap pseudoreplicates [26]. For better visualization, the tree was edited using iTol website (v6.4.2) [27].

## Statistical analysis

We tested for differences in continuous baseline clinical characteristics and outcomes by MRSA status using the Wilcoxon rank sum test. The Wilcoxon test is a nonparametric test that is robust to departures from Normality, so we did not evaluate the data distribution prior to testing. Differences in categorical variables by MRSA status, severity score, or time-period were tested using Pearson's chi-squared test. For categorical variables with more than two levels, we report the P-value testing the null hypothesis that all proportions are equal versus the alternative hypothesis that at least one proportion differs.

## Results

### Patient demographics and clinical outcomes

We enrolled 118 children with invasive *S. aureus* infections who met eligibility criteria for the analysis. Whole genome sequencing was performed on 119 clinical isolates, and full clinical characteristics for 118 children were evaluated (one patient with invasive infection had 2 distinct *S. aureus* isolates). The median age at enrollment was 9.2 years, with similar

ages between children with MRSA (8.9 years) and MSSA (9.3 years) infections (Table 1). The majority of participants (65%) were male. There were no significant differences in ethnicity or race with regard to risk of MRSA vs. MSSA invasive infection. There was 1 death in this cohort, which was caused by MSSA. Overall, MSSA invasive infection was more frequent than MRSA (56% vs. 44%). MRSA was less frequent in the latter half of the time period, decreasing from 59% from 2010–2014 to 42% from 2019–2022 (Table 2).

Baseline clinical data and complications are summarized in Table 3. The most frequent clinical phenotype was acute hematogenous osteomyelitis (42%), followed by myositis (12%). Infection with MRSA, compared to MSSA, was associated with greater inflammation early in the disease course, including increased white blood cell (WBC) count (14,000/mcL vs 12,100/mcL, p = 0.03), higher absolute neutrophil count at admission (10,600/mcL vs 9,000/mcL, p = 0.02), higher peak WBC count (14,800/mcL vs 13,100/mcL, p = 0.05), and higher peak CRP (193 mg/L vs 151 mg/L, p = 0.05). However, clinical outcomes and disease complications, including hospital length of stay, ICU admission, days in ICU, septic emboli, and prolonged bacteremia, did not differ between MRSA and MSSA invasive infection.

## Molecular and genotypic characteristics

There were 9 unique clonal complexes (CCs) identified among invasive and colonization isolates: CC1, CC5, CC8, CC15, CC22, CC30, CC45, CC97, CC121. Invasive Isolates from 2014–2022 exhibited higher clonal complex diversity than invasive isolates from 2010–2012. Greater clonal complex diversity was seen in colonization isolates compared to invasive isolates (Fig 1). The genes encoding surface protein staphylokinase (Sak) and the leukocidin LukED were strongly associated with invasive isolates compared to colonization isolates, a finding that persisted over time (Fig 2).

**Table 1. Study Demographics.**

| N = 118 | MRSA (n = 52) | MSSA (n = 66) | Combined |
|---|---|---|---|
| Age in years, median (IQR) | 8.9 (6.3-11.3) | 9.3 (6.1-13.2) | 9.2 (6.1-12.5) |
| **Gender** | | | |
| Female | 33% | 36% | 35% |
| Male | 67% | 64% | 65% |
| **Race/Ethnicity** | | | |
| American Indian/Alaska Native | 0% | 2% | 1% |
| Black or African American | 17% | 23% | 20% |
| White | 81% | 74% | 77% |
| Unknown/Not Reported | 0% | 2% | 1% |
| Other | 2% | 0% | 1% |
| Hispanic or Latino | 2% | 3% | 3% |
| NOT Hispanic or Latino | 77% | 91% | 85% |
| Unknown/Not Reported | 21% | 6% | 13% |

**Table 2. Antibiotic resistance over time.**

| N = 118 | 2010-2014 n = 44 | 2015-2018 n = 48 | 2019-2022 n = 26 | P-value |
|---|---|---|---|---|
| Clindamycin | 16% | 12% | 15% | 0.89 |
| TMP/SMX | 2% | 4% | 0% | 0.55 |
| Erythromycin | 66% | 44% | 38% | 0.04 |
| Doxycycline | 0% | 0% | 0% | – |
| Meth/Oxa (MRSA) | 59% | 31% | 42% | 0.03 |

**Table 3. Clinical Data and Complications.**

| Infection Type | N = 118 | MRSA (52) | MSSA (66) | Combined | |
|---|---|---|---|---|---|
| Osteomyelitis | | 48% | 52% | 42% (50/118) | |
| Septic arthritis | | 25% | 75% | 3% (4/118) | |
| Multifocal musculoskeletal infection | | 44% | 56% | 8% (9/118) | |
| Myositis | | 36% | 64% | 12% (14/118) | |
| Pneumonia | | 60% | 40% | 4% (5/118) | |
| Endocarditis | | 25% | 75% | 3% (4/118) | |
| Primary Bacteremia/Sepsis | | 33% | 67% | 5% (6/118) | |
| Osteomyelitis and Septic arthritis | | 50% | 10% | 9% (10/118) | |
| Other | | 44% | 56% | 14% (16/118) | |
| **Baseline Clinical Data** | **N** | **MRSA** | **MSSA** | **Combined** | **p-value** |
| Admission WBC count (k/mcL) | 113 | 14.0 (11.0-17.6) | 12.1 (8.6-16.0) | 12.8 (9.5-17.0) | 0.03 |
| Peak WBC Count (k/mcL) | 118 | 14.8 (11.1-20.2) | 13.1 (9.2-17.9) | 13.7 (10.5-19.0) | 0.05 |
| Nadir WBC Count (k/mcL) | 118 | 9.4 (7.6-11.2) | 8.4 (6.8-10.5) | 8.6 (6.9-10.9) | 0.18 |
| Admission ESR (mm/hr) | 95 | 57 (27-81) | 53 (38-69) | 53 (36-79) | 0.91 |
| Peak ESR (mm/hr) | 103 | 76 (49-97) | 59 (38-69) | 63 (46-88) | 0.07 |
| Admission CRP (mg/L) | 100 | 140 (87-246) | 108 (62-176) | 113 (74-194) | 0.13 |
| Peak CRP (mg/L) | 111 | 193 (101-274) | 151 (86-205) | 164 (94-251) | 0.05 |
| Admission platelet count (k/mcL) | 113 | 272 (224-368) | 238 (190-355) | 253 (201-360) | 0.11 |
| Nadir platelet count (k/mcL) | 118 | 270 (198-356) | 222 (184-343) | 240 (188-350) | 0.17 |
| Admission ANC (k/mcL) | 109 | 10.6 (7.9-13.7) | 9.0 (5.8-12.4) | 10.0 (6.5-12.9) | 0.02 |
| Peak ANC (k/mcL) | 115 | 10.6 (8.3-15.9) | 9.3 (6.4-13.2) | 10.3 (7.0-15.0) | 0.11 |
| Nadir ANC (k/mcL) | 115 | 5.9 (4.3-7.3) | 4.9 (3.3-7.0) | 5.2 (3.6-7.1) | 0.19 |
| Peak Temperature (°F) | 118 | 103 (101-103) | 103 (102-103) | 103 (101-103) | 0.50 |
| **Clinical Complications** | | **MRSA** | **MSSA** | **Combined** | **p-value** |
| Hospital length of stay (days) | 118 | 9 (7-14) | 8 (6-10) | 8 (6-12) | 0.15 |
| ICU Admission | 118 | 17% | 14% | 15% | 0.58 |
| Days in ICU | 18 | 4.5 (3-5.5) | 11 (4-21) | 5 (3-11) | 0.11 |
| **Complications** | | | | | |
| Thrombosis/septic emboli | 118 | 13% | 12% | 13% | 0.83 |
| Death | 118 | 0% | 2% | 1% | 0.37 |
| Prolonged bacteremia | 118 | 37% | 27% | 31% | 0.28 |
| **Severity Score** | 108 | **MRSA** | **MSSA** | **Combined** | **p-value: 0.24** |
| 0 | 33 | 27% | 33% | 31% | |
| 1 | 34 | 27% | 35% | 31% | |
| 2 | 19 | 17% | 18% | 18% | |
| 3 or more | 22 | 29% | 13% | 20% | |

Staphylokinase was detected in 91% (107/118) of invasive isolates and 71% (175/246) of colonization isolates (p < 0.001), and in all clonal complexes except CC15. LukED was detected in 80% (94/118) of invasive isolates and 63% (156/246) of colonization isolates (p = 0.002). LukED was detected in 95% (20/21) of patients with a severity score of 3 or more (Table 4). Conversely, the genes encoding toxic shock syndrome toxin-1 (TST) and staphylococcal enterotoxin B (SEB) were found more commonly in colonization isolates (11–25%) compared to invasive isolates (2–9%). The genes encoding leu-kocidin AB (LukAB), alpha-hemolysin (Hla), iron-regulated surface proteins A and B (IsdA, IsdB), staphylococcal binding

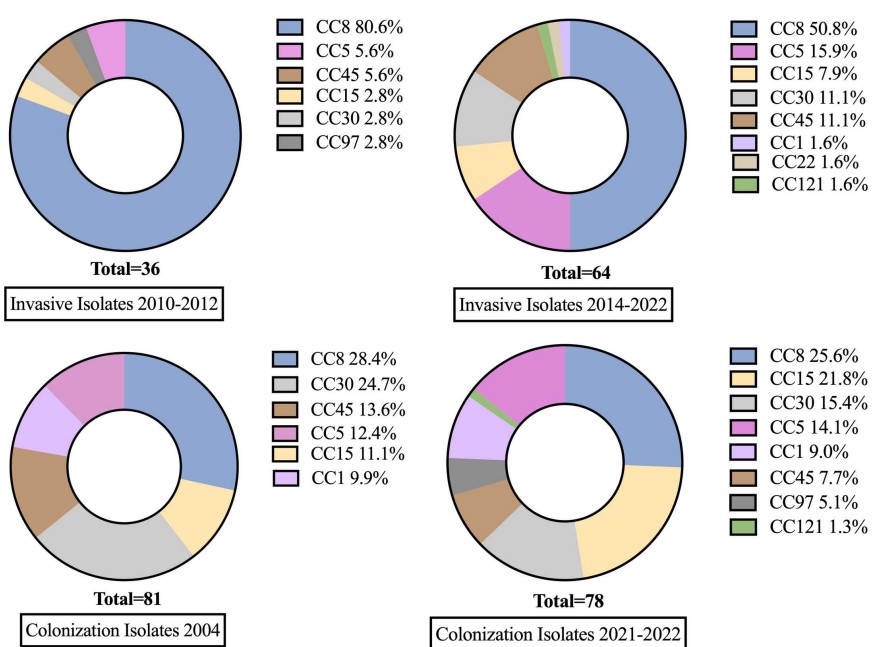

**Fig 1. Clonal complex composition of colonization and invasive cohorts.** Clonal Complex composition of invasive and colonizing cohorts.

immunoglobulin protein (sbi), extracellular fibrinogen-binding protein (Efb), and clumping factors A and B (ClfA, ClfB) were present in all isolates.

CC8 remains the most prevalent invasive clonal complex, though its prevalence has decreased significantly over time (67% [29/43] in 2010–2012 vs. 43% [32/75] in 2014–2022, p = 0.001). The USA300 diagnostic alleles analysis revealed that only 12 genomes were classified as USA300. Concomitant with the decline over time of CC8 was an increased frequency of invasive infection caused by CC5, CC45, CC30, and CC15 strains. CC1, CC22, and CC121 emerged as new causes of severe invasive infection in this cohort. Greater clonal complex diversity was observed in invasive MSSA isolates compared to MRSA (Fig 3).

The prevalence of key virulence genes in clinically invasive *S. aureus* isolates also changed significantly over time (Fig 4). The frequency of accessory gene regulator type 1 (agr 1), Panton-Valentine Leukocidin (PVL), enterotoxins K and Q (SEK/SEQ), and the arginine catabolic mobile element (ACME) decreased significantly in *S. aureus* isolates since 2010 (p < 0.0001). The decline of these virulence factors correlates with the decline of CC8; PVL, agr type 1, SEK/SEQ, and ACME were found in 98% (60/61) 82% (50/61), 69% (42/61) and 61% (37/61) of CC8 invasive isolates, respectively. Diversity of agr type increased as clonal complex diversity increased over time (Table 5). Toxin carriage was evaluated between MRSA and MSSA isolates. SEB and TST prevalence did not differ significantly between MRSA and MSSA. LukED and PVL were more prevalent in MRSA isolates (Table 6).

Colonization and invasive strain relatedness are displayed on a phylogenetic tree, and sequence type was identified (Fig 5). All isolates belonging to ST1, ST109, ST398, ST1290 were colonization isolates. The isolate belonging to ST22 was an invasive isolate. All other sequence types included both colonization and invasive isolates.

Phenotypic antimicrobial resistance was evaluated (Tables 7 and 8). There was no correlation between antibiotic resistance and clinical severity score. Resistance to clindamycin or trimethoprim-sulfamethoxazole was not significantly different between MRSA and MSSA isolates, though erythromycin resistance was more common in MRSA than MSSA isolates (85% vs 24%, p < 0.001). Doxycycline resistance was not detected in any isolates.

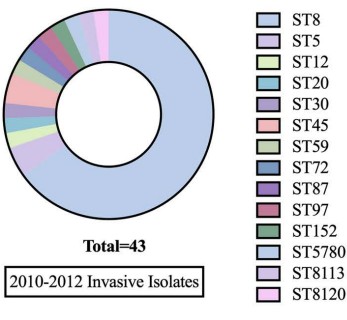

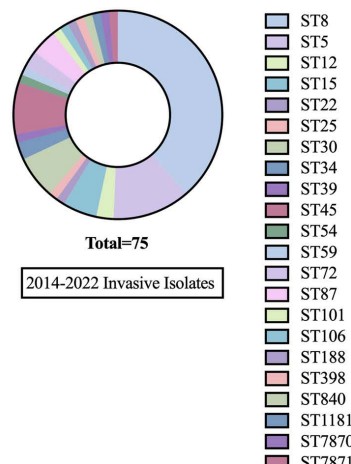

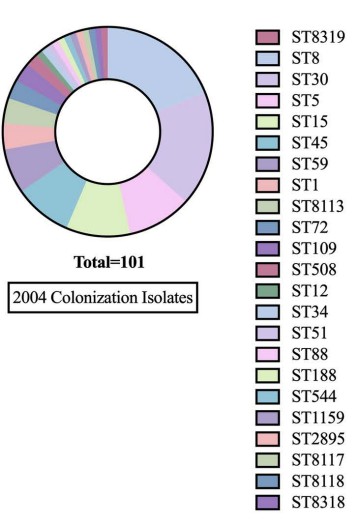

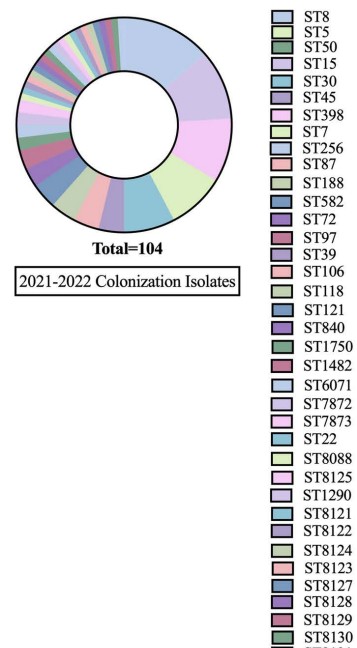

**Fig 2. Sequence type diversity in invasive and colonization cohorts.** Sequence type diversity increased in invasive isolates over time. Sequence type diversity was greater in colonization isolates compared to invasive isolates.

## Discussion

In this cohort of pediatric patients with invasive *S. aureus* infections, the clinical severity of invasive infections with MRSA vs. MSSA infections was similar. While invasive MRSA infections were associated with more inflammation early in the course of illness, there were no statistically significant differences in clinical complications or outcomes between MRSA and MSSA. Importantly, we observed a significant expansion in the diversity of invasive isolates and shifts in the presence of virulence genes associated with invasive infection.

**Table 4. Micro and molecular epi descriptive statistics by severity score.**

| | N | 0<br>N = 34 | 1<br>N = 34 | 2<br>N = 19 | 3 or more<br>N = 21 | P-value |
|---|---|---|---|---|---|---|
| SEB | 118 | 3% | 6% | 11% | 5% | 0.71 |
| HlgB | 118 | 18% | 26% | 16% | 5% | 0.23 |
| TST | 118 | 6% | 6% | 0% | 5% | 0.76 |
| LukED | 118 | 88% | 68% | 74% | 95% | 0.039 |
| PVL | 118 | 41% | 44% | 53% | 67% | 0.28 |
| USA type 300 | 116 | 29% | 24% | 33% | 57% | 0.071 |
| Sequence type | 117 | | | | | 0.14 |
| ST30 | | 3% | 6% | 5% | 0% | |
| ST45 | | 9% | 12% | 5% | 0% | |
| ST5 | | 21% | 9% | 0% | 14% | |
| ST72 | | 3% | 6% | 0% | 0% | |
| ST8 | | 39% | 32% | 58% | 76% | |
| ST87 | | 0% | 6% | 11% | 0% | |
| Clonal Complex | 94 | | | | | 0.21 |
| CC30 | | 8% | 11% | 13% | 0% | |
| CC45 | | 12% | 18% | 7% | 0% | |
| CC5 | | 21% | 7% | 0% | 18% | |
| CC8 | | 54% | 50% | 67% | 82% | |

The increase in MSSA is accompanied by an increase in the diversity of sequence types associated with invasive infection. CC1, CC5, CC8, CC15, CC30, CC45, CC97, and CC121 were found in invasive and colonization isolates, with the emergence of CC1, CC22 and CC121 in recent years. These findings build upon prior work demonstrating that these clonal complexes can cause a wide spectrum of infections and that colonization typically is a prerequisite for infection [28–31]. Ninety-three percent (14/15) of CC1 isolates were colonization isolates; others have also observed that CC1 is a rare cause of invasive infection [28,32,33].

This study has potential implications for infection prevention and control. Inpatient infection prevention efforts surrounding *S. aureus* are challenging, in part because the pathogen-specific determinants of invasive *S. aureus* infection are unknown. The rise of invasive MSSA is accompanied by diversification in clonal complex distribution. Because both MSSA and MRSA cause invasive *S. aureus* infection in the NICU with similar incidence and outcomes [34,35], the luxury of equating pathogenicity with methicillin resistance no longer exists.

Some virulence genes declined over time, while others remained persistently elevated in either invasive or colonizing cohorts. The decline of virulence genes previously associated with invasive infection in children, such as PVL, SEK/SEQ, and ACME, can largely be attributed to the decline in prevalence of the CC8 *S. aureus* lineage [36–38]. Virulence genes align with clonal complex when vertical transfer occurs, though virulence genes spread across clonal complexes when horizontal transfer transpires, optimizing bacterial fitness [39]. Staphylokinase, found in all clonal complexes except CC15 in this study, was significantly more prevalent in invasive isolates compared to colonization isolates. Sak is located on a prophage that contains an immune evasion cluster, and its presence on a prophage could allow for potential spread from invasive to commensal populations. Wang et al found an association between staphylokinase and invasive human infection when comparing rates of staphylokinase carriage in livestock-associated *S. aureus* to human-adapted staph aureus. They also observed milder pneumonia in mice infected with sak gene knockout isolates [40]. Staphylokinase is a thrombolytic enzyme that enables *S. aureus* dissemination in the setting of abscess formation [41]; its primary mode of action is converting plasminogen to plasmin (a genetically modified version of staphylokinase is used clinically for thrombolysis in

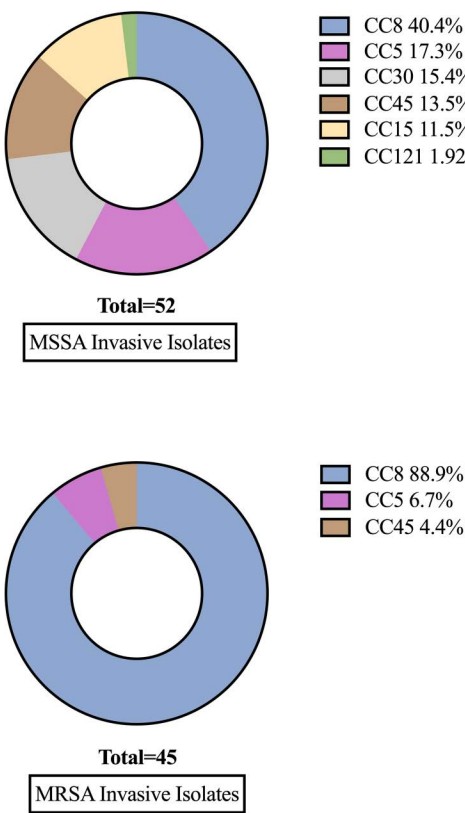

**Fig 3. Clonal complex composition of MSSA and MRSA invasive isolates.**

patients with acute myocardial infarction or stroke [42]), though it also enhances immune evasion by binding C3b and the Fc portion of immunoglobulin [43]. Given its role in immune evasion and pathogenesis, it may be reasonable to consider staphylokinase as a target in next-generation staphylococcal interventions [44]. LukED was also more frequently identified in invasive isolates. LukED is a bicomponent pore-forming leukocidin that lyses erythrocytes to facilitate iron acquisition [45]. Spaan et al have demonstrated that LukED and HlgAB target the Duffy antigen receptor for chemokines (DARC) but use different mechanisms to release iron from erythrocytes [46]. While HlgAB has been found in over 99% of *S. aureus* isolates infecting humans [47], LukED is found in approximately 80% of *S. aureus* isolates infecting humans [48]. Whether *S. aureus* has additional virulence factors that function to lyse erythrocytes is an area requiring further research. The higher prevalence of TST and SEB in the colonization cohorts is consistent with their known roles in toxin-mediated disease rather than invasive infection [49,50].

While multiple epidemiologic studies have demonstrated the increase in MSSA among pediatric patients with *S. aureus* infections, the exact reason for MSSA's resurgence is unclear [6,7,51]. It has been shown that MSSA and MRSA compete for colonization space [52]; therefore, it is reasonable to hypothesize that increased efforts to combat MRSA transmission in the healthcare setting has led to a relative increase in MSSA. Some of the increased diversity may be attributable to animal acquisition. ST97 and ST398, clinically relevant strains that were both detected in this study, are strongly associated with livestock exposure [53].

Our study should be interpreted in the context of several limitations. First, colonization isolates in this study were obtained from different populations (colonization isolates from 2004 were obtained from healthy pediatric patients in the outpatient setting, and colonization isolates from 2021–2022 were obtained from patients admitted to the neonatal

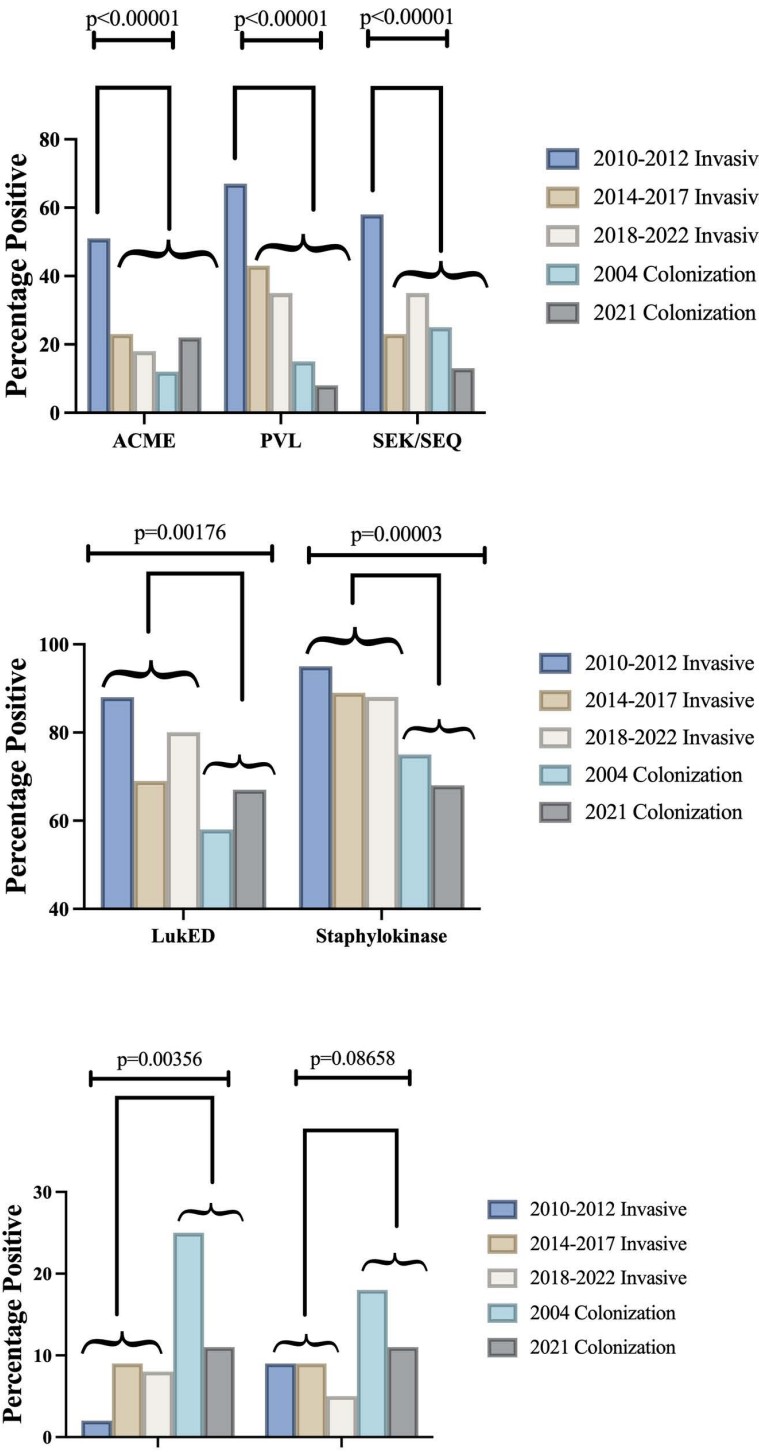

**Fig 4. Virulence factor prevalence in invasive and colonization isolates.** ACME, PVL, and SEK/SEQ declined over time. LukED and Staphylokinase were highly prevalent in invasive isolates over time compared to colonization. TST and SEB were more prevalent in colonization isolates compared to invasive isolates.

**Table 5. Agr gene carriage.**

| | N = 118 | Combined | MRSA | MSSA |
|---|---|---|---|---|
| Agr 1 | | 73% | 90% | 59% |
| 2 | | 18% | 10% | 24% |
| 3 | | 7% | 0% | 14% |
| 4 | | 2% | 0% | 3% |
| **Trends over time** | | | | |
| | N | 2010-2014 n = 44 | 2015-2018 n = 48 | 2019-2022 n = 26 |
| Agr 1 | 118 | 89% | 62% | 65% |
| 2 | | 11% | 21% | 23% |
| 3 | | 0% | 15% | 8% |
| 4 | | 0% | 2% | 4% |

**Table 6. Toxin carriage between MRSA and MSSA (N = 118).**

| | Total | MRSA N = 52 | MSSA N = 66 | P-value |
|---|---|---|---|---|
| SEB | 5% | 2% | 8% | 0.17 |
| Hla | 100% | 100% | 100% | – |
| TST | 5% | 2% | 8% | 0.17 |
| LukAB | 100% | 100% | 100% | – |
| LukED | 82% | 90% | 76% | 0.039 |
| PVL | 48% | 77% | 26% | <0.001 |

intensive care unit). Despite their different sources, most isolates in both colonization cohorts were MSSA (72% in 2004 and 76% in 2021–2022), and most belonged to CC5, CC8, CC15, CC30, or CC45. Second, patients in the invasive infection cohort were not tested for colonization because patients in this group had received at least 48 hours of antibiotics prior to an invasive culture becoming positive; we opted not to test these patients for colonization as we anticipated a high rate of false negatives. Third, this was a single center study in the southern United States. As all isolates were collected at Vanderbilt University Medical Center in Tennessee, our observations in phylogenetic drift may not be generalizable to areas outside this geographic area. Despite enrolling over 100 patients with invasive *Staphylococcus aureus* infection, the study was not powered to detect subtle changes in clinical outcomes. A large, multicenter study would be required to detect more nuanced differences.

Significant shifts have occurred over the past decade in the predominant circulating *S. aureus* strains and their virulence factor repertoires. The once-dominant CC8 (USA300 clone) has receded; concomitantly, there has been a significant reduction in the prevalence of virulence genes once thought to be crucial for pathogenesis. While PVL, ACME, agr type 1 and SEK/SEQ decreased over time, we found that the frequency of staphylokinase and LukED was persistent and remained significantly higher in invasive isolates compared to colonization isolates. These findings have implications for vaccine development, as vaccines should target virulence factors that are found consistently over time. These findings also have implications for infection prevention and control practices, as eradication of colonizing strains with lower potential for invasive infection may inadvertently allow for replacement by more virulent strains. Given the significant global burden of *S. aureus* infection, further research is necessary to understand the dynamics of transmission and progression of disease.

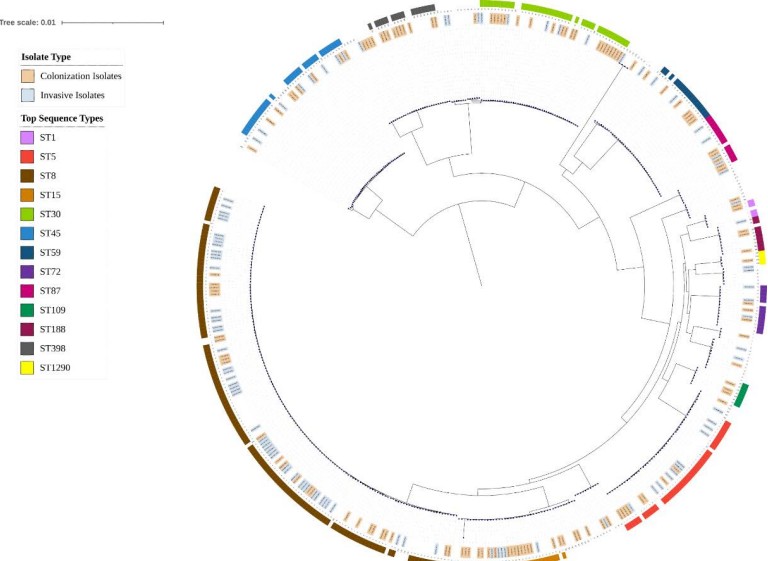

**Fig 5. Phylogenetic tree of colonization isolates (orange) and invasive isolates (light blue).** All isolates belonging to ST1, ST109, ST398, and ST1290 were colonization isolates. All other sequence types included both colonization and invasive isolates.

**Table 7. Antibiotic resistance.**

| N = 118 | MRSA | MSSA | P-value |
|---|---|---|---|
| Clindamycin | 17% | 12% | 0.43 |
| TMP/SMX | 0% | 5% | 0.12 |
| Erythromycin | 85% | 24% | <0.001 |
| Doxycycline | 0% | 0% | – |

**Table 8. Antibiotic resistance and severity score.**

| N = 118 | 0<br>n = 48 | 1<br>n = 48 | 2<br>n = 19 | 3 or more<br>n = 21 | p-value |
|---|---|---|---|---|---|
| Clindamycin | 21% | 15% | 5% | 14% | 0.52 |
| TMP/SMX | 6% | 3% | 0% | 0% | 0.50 |
| Erythromycin | 44% | 44% | 63% | 62% | 0.33 |
| Doxycycline | 0% | 0% | 0% | 0% | – |
| Meth/Oxa | 41% | 38% | 42% | 62% | 0.35 |

## Supporting information

**S1 File. Virulence factors Please see supporting file "S1 File" for genomic determination of virulence factors.** (XLSX)

**S2 File. De-identified consensus sequences submitted to PubMLST Sequence typing data are publicly available via PubMLST. Please see supporting file "S2 File."** (XLSX)

## Acknowledgments

We acknowledge Janet Shelton and Suzanne Miskel for administrative assistance; Angela Jones for technical assistance.

## Author contributions

**Conceptualization:** Margaret Free, Isaac Thomsen.

**Data curation:** Margaret Free, Nicole Soper, Andries Feder.

**Formal analysis:** James C. Slaughter, Andries Feder, Colleen Bianco.

**Funding acquisition:** Margaret Free, Isaac Thomsen.

**Investigation:** Margaret Free, Nicole Soper, C. Buddy Creech, Isaac Thomsen.

**Methodology:** Paul Planet.

**Resources:** Margaret Free, Paul Planet, Isaac Thomsen.

**Software:** Andries Feder, Colleen Bianco, Ahmed M. Moustafa.

**Supervision:** Paul Planet, C. Buddy Creech, Isaac Thomsen.

**Visualization:** Margaret Free.

**Writing – original draft:** Margaret Free.

**Writing – review & editing:** Nicole Soper, James C. Slaughter, Andries Feder, Colleen Bianco, Ahmed M. Moustafa, Paul Planet, C. Buddy Creech, Isaac Thomsen.

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
