## [Decision Letter · Decision Letter 0]

PONE-D-24-08440Evolution of strain diversity and virulence factor repertoire in pediatric Staphylococcus aureus isolatesPLOS ONE

Dear Dr. Free,

Thank you for submitting your manuscript to PLOS ONE. After careful consideration, we feel that it has merit but does not fully meet PLOS ONE’s publication criteria as it currently stands. Therefore, we invite you to submit a revised version of the manuscript that addresses the points raised during the review process. Please submit your revised manuscript by Jun 22 2024 11:59PM. If you will need more time than this to complete your revisions, please reply to this message or contact the journal office at plosone@plos.org . Please include the following items when submitting your revised manuscript:

We look forward to receiving your revised manuscript.

Kind regards,

Nagendra N. Mishra, Ph.D

Academic Editor

PLOS ONE

Journal Requirements:

This work was supported through the NIH/NIAID grants 5R01AI139172-02 (Thomsen PI) and 5T32AI095202 (Childhood Infections Research Program Training Grant). The funders had no role in study design, data collection and interpretation, or submission of the work for publication. The data used in Fig 3 were collected through the MENDEL high performance computing (HPC) cluster at the American Museum of Natural History. This HPC cluster was developed with National Science Foundation (NSF) Campus Cyberinfrastructure support through Award#1925590. We thank the VUMC Clinical Microbiology Laboratory, the Vanderbilt Vaccine Research Program Laboratory (VVRP), and the Vanderbilt Technologies for Advanced Genomics (VANTAGE) Core. VANTAGE is supported in part by Clinical and Translational Science Award Grant 5UL1 RR024975-03, Vanderbilt Ingram Cancer Center Grant P30 CA68485, Vanderbilt Vision Center Grant P30 EY08126, and National Institutes of Health/National Center for Research Resources Grant G20 RR030956. 

5. We note that your Data Availability Statement is currently as follows: All relevant data are within the manuscript and its Supporting Information files.

Reviewers' comments:

Reviewer's Responses to Questions

**Comments to the Author**

1. Is the manuscript technically sound, and do the data support the conclusions?

Reviewer #1: Yes

Reviewer #2: Yes

Reviewer #3: Partly

2. Has the statistical analysis been performed appropriately and rigorously? 

Reviewer #1: Yes

Reviewer #2: Yes

Reviewer #3: Yes

3. Have the authors made all data underlying the findings in their manuscript fully available?

Reviewer #1: Yes

Reviewer #2: Yes

Reviewer #3: Yes

4. Is the manuscript presented in an intelligible fashion and written in standard English?

Reviewer #1: Yes

Reviewer #2: Yes

Reviewer #3: Yes

5. Review Comments to the Author

**Reviewer #1:**  This is an interesting assessment of pediatric S aureus isolates (MSSA and MRSA) at a single academic medical center. The authors reveal some changes in susceptibility (loss of macrolide efflux with increased susceptibility over time with clinda staying constant) and shift toward the 'big 2' CC8 and CC8 being responsibile for more invasive infections.

While a genotypic analysis is helpful, expression of some key virulence exotoxins (eg alpha toxin) would be of interest. For example, while the strains all contain the alpha toxin gene, I suspect toxin expression will be variable over time . One might anticipate community-based infection to harbor more alpha-toxin producing strains while nosocomial strains might not express . For example, there is less need for exotoxin-driven invasion in the hospital when health care delivery has already granted easy access via wounds and catheters. The parsimony of evolution would be expected to relinquish the metabolic demand for such toxins in these settings.

The expression of alpha toxin can actually be done grossly through a crude hemolysis assay using beta-toxin disks

Minor comment-- Why were the catheter and foreign body infection isolates not sent for sequencing?

**Reviewer #2:**  I read with interest the contribution by Free et al. (PONE-D-24-08440). The authors describe the phylogenetic drift in invasive pediatric Staphylococcus aureus isolates over the past decade. The manuscript was a pleasure to read with very clear writing. Its findings are well founded and introduced with the appropriate context of existing staphylococcal phylogeny knowledge. While I may question the use of neonatal isolates as a control to represent commensal isolates across the pediatric age spectrum, the authors clearly define this as a study limitation. All of my comments are minor and reflective of the superior skill of the authors to communicate their scientific findings. Of these, the most significant in my opinion would be to clarify the single-center/multi-center nature of the sample collection and, if multi-center, how many isolates were recovered from each site. Thank you for the opportunity to review this strong submission for publication with PLOSone. I would be happy to review any future revisions.

Identified Areas for Improvement:

* In your conclusions section you state that sak and lukED were associated with invasive infection over time. What exactly do you mean by this? Based on your figures, sak and lukED were clearly associated with invasion (consistent with observations in the adult population) but I don’t see a clear temporal relationship (if anything, the prevalence of sak and lukED positivity appears to go down with time).

* A recent study specifically identified sak as the genetic element most predictive of invasion and poor clinical outcome. In addition to including this reference, it might add interest to note that sak is typically localized to a prophage, specifically one that also bears an immune evasion cluster, is suspected to be easy to mobilize and potentially able to persist as an episome. This would conceivably facilitate its dissemination between populations and conversion of commensal populations into invasive ones.

* Please modify the date range for column 2 in table 2 for accuracy.

* Please comment on the breakdown into different date ranges. Why does the first cohort span six years, the second five and the third only four? Was there a reason for the differences?

* Please comment on the geographical distribution of your collection. Were all pediatric cases collected at CHOP? VUMC? If it is multi-center, that may enrich your phylogenetic distribution, although your controls seem to have been only collected at a single center. Regardless, a more robust discussion of the study sites and their strengths/limitations is warranted. I would also consider a statement about the external validity of your study – the results are representative of isolates from the Eastern United States; I’m not certain they would represent pediatric isolates in other areas of the world.

* Regarding graphs and figures, please consider a color scheme that allows for differentiation when printed in black-and-white.

* I may have missed this, but if you haven’t submitted your sequence reads, contigs or assembled files to NCBI or EMBO, please consider doing so to more comprehensively satisfy the data availability requirements.

**Reviewer #3:**  This work describes the genetic evolution of S. aureus in pediatrics over the last 12 years (2010-2022). The methods are technically sound and well described, albeit the comparison of invasive infection vs colonization with different timeframes is weak. The findings from this study indicate a transition of S. aureus clonal type and pathogenicity over this time period. These data are new to the field as ongoing S. aureus genetic epidemiology is unique to the given situation. There are areas of the manuscript that require some attention including

1) The colonization timeline comparison (vs invasive infection) is difficult to justify and therefore make general comparisons. Some clarification/further details as to why the colonization cohorts were selected and different than then invasive group is needed.

2) There is good evidence that patients who are colonized with S. aureus and become infected have less severe acute infections. Were the patients from the invasive infection group tested for colonization as well? If yes, were they included or excluded? Further details on this as well as discussion of these known factors is warranted.

3) The authors should provide a definition for severe immunocompromised as that is an exclusion factor in this study.

4) It is odd that mortality is not included in the disease severity. Although only 1 patient succumbed to mortality in the study, was this patient deemed severe?

5) For statistical analysis, the authors used Wilcoxon rank test as one of the tests. There is no indication that the data are nonparametric distribution or evaluated for data distribution for statistical testing.

6) The range of data is not presented in some of the demographic data and tables. See table 3 for example.

7) The authors should indicate comparisons for the P values in table 2. There are 3 time periods presented and P values for each.

8) Table 2 shows a column of 2019-2014. This appears to be an error and should be corrected.

9) There are some additional susceptibility profiles that would be valuable to add a) vancomycin MIC in MRSA and MSSA as this has trended down over in the last 10+years and b) penicillin MIC in MSSA as evidence is emerging that S. aureus is becoming more penicillin susceptible. Both antibiotics would be valuable to include.

10) In the discussion, Lines 319-330 are highly speculative. The low number of isolates limits the extrapolation of the data to their hypotheses. It seems that the authors may be trying to fit their data to known assumptions without listing limitations

11) There is discussion of staphylokinase as a potential target. However, how do the authors propose targeting staphylokinase when the clinical relevance is incompletely defined (Line 339)? These statements seem to contradict.

6. PLOS authors have the option to publish the peer review history of their article (what does this mean? ). If published, this will include your full peer review and any attached files.

**Do you want your identity to be public for this peer review?** For information about this choice, including consent withdrawal, please see our Privacy Policy .

Reviewer #1: No

Reviewer #2: **Yes: ** Andrew David Berti

Reviewer #3: No

---

## [Author Response · Author response to Decision Letter 1]

1 Nov 2024

All relevant data are within the manuscript, its Supporting Information files (Virulence factor sheet and Sequences Submitted to PubMLST), and from the PubMLST database (https://pubmlst.org/bigsdb?db=pubmlst_saureus_isolates).

Values behind the means have been added to table 3. The following sentence has been added to the manuscript: “Consensus sequences with MLST assignments remain publicly available via PubMLST.” De-identified consensus sequence names corresponding to bacterial isolates have been uploaded under the document entitled “Sequences submitted to PubMLST.”

---

## [Decision Letter · Decision Letter 1]

PONE-D-24-08440R1Evolution of strain diversity and virulence factor repertoire in pediatric Staphylococcus aureus isolatesPLOS ONE

Dear Dr. Free

Thank you for submitting your manuscript to PLOS ONE. After careful consideration, we feel that it has merit   but does not fully meet PLOS ONE’s publication criteria as it currently stands. Therefore,  I am considering the potential acceptance of the manuscript and  invite you to submit a revised version of the manuscript that addresses the points raised during the review process by reviewer 3. 

Please submit your revised manuscript by Jan 29 2025 11:59PM. If you will need more time than this to complete your revisions, please reply to this message or contact the journal office at plosone@plos.org . Please include the following items when submitting your revised manuscript:

We look forward to receiving your revised manuscript.

Kind regards,

Nagendra N. Mishra, Ph.D

Academic Editor

PLOS ONE

Reviewers' comments:

Reviewer's Responses to Questions

**Comments to the Author**

1. If the authors have adequately addressed your comments raised in a previous round of review and you feel that this manuscript is now acceptable for publication, you may indicate that here to bypass the “Comments to the Author” section, enter your conflict of interest statement in the “Confidential to Editor” section, and submit your "Accept" recommendation.

Reviewer #2: All comments have been addressed

Reviewer #3: (No Response)

2. Is the manuscript technically sound, and do the data support the conclusions?

Reviewer #2: Yes

Reviewer #3: Yes

3. Has the statistical analysis been performed appropriately and rigorously? 

Reviewer #2: Yes

Reviewer #3: Yes

4. Have the authors made all data underlying the findings in their manuscript fully available?

Reviewer #2: Yes

Reviewer #3: Yes

5. Is the manuscript presented in an intelligible fashion and written in standard English?

Reviewer #2: Yes

Reviewer #3: Yes

6. Review Comments to the Author

Reviewer #2: (No Response)

Reviewer #3: The revision does not appear to address or rebut any comments provided on the prior review. For example, expression of toxin was not addressed, susceptibility of interest was not addressed and many others.

7. PLOS authors have the option to publish the peer review history of their article (what does this mean? ). If published, this will include your full peer review and any attached files.

**Do you want your identity to be public for this peer review?** For information about this choice, including consent withdrawal, please see our Privacy Policy .

Reviewer #2: **Yes: ** Andrew David Berti

Reviewer #3: No

---

## [Author Response · Author response to Decision Letter 2]

24 May 2025

I greatly appreciate the thoughtful comments from the reviewers. I have addressed their comments in the "Response to Reviewers" document.

---

## [Editor Report · Decision Letter 2]

Evolution of strain diversity and virulence factor repertoire in pediatric Staphylococcus aureus isolates

PONE-D-24-08440R2

Dear Dr. Free

We’re pleased to inform you that your manuscript has been judged scientifically suitable for publication and will be formally accepted for publication once it meets all outstanding technical requirements.

Kind regards,

Nagendra N. Mishra, Ph.D

Academic Editor

PLOS ONE
---

## [Editor Report · Acceptance letter]

PONE-D-24-08440R2

PLOS ONE

Dear Dr. Free,

I'm pleased to inform you that your manuscript has been deemed suitable for publication in PLOS ONE. Congratulations! Your manuscript is now being handed over to our production team.

Kind regards,

on behalf of

Dr. Nagendra N. Mishra

Academic Editor

PLOS ONE